# Dynamic Mechanical and Creep Behaviour of Meltspun PVDF Nanocomposite Fibers

**DOI:** 10.3390/nano11082153

**Published:** 2021-08-23

**Authors:** Fatemeh Mokhtari, Geoffrey M. Spinks, Sepidar Sayyar, Javad Foroughi

**Affiliations:** 1Intelligent Polymer Research Institute, University of Wollongong, Wollongong, NSW 2500, Australia; fatemehm@uow.edu.au (F.M.); gspinks@uow.edu.au (G.M.S.); sepidar@uow.edu.au (S.S.); 2School of Electrical, Computer and Telecommunications Engineering, Faculty of Engineering and Information Sciences, University of Wollongong, Wollongong, NSW 2522, Australia; 3Westgerman Heart and Vascular Center, University of Duisburg-Essen, 45122 Essen, Germany

**Keywords:** polyvinylidene fluoride (PVDF), composite fibers, piezoelectric, dynamic mechanical analysis, creep, storage modulus

## Abstract

Piezoelectric fibers have an important role in wearable technology as energy generators and sensors. A series of hybrid nanocomposite piezoelectric fibers of polyinylidene fluoride (PVDF) loaded with barium–titanium oxide (BT) and reduced graphene oxide (rGO) were prepared via the melt spinning method. Our previous studies show that high-performance fibers with 84% of the electroactive β-phase in the PVDF generated a peak output voltage up to 1.3 V and a power density of 3 W kg^−1^. Herein, the dynamic mechanical and creep behavior of these fibers were investigated to evaluate their durability and piezoelectric performance. Dynamic mechanical analysis (DMA) was used to provide phenomenological information regarding the viscoelastic properties of the fibers in the longitudinal direction. DSC and SEM were employed to characterize the crystalline structure of the samples. The storage modulus and the loss tangent increased by increasing the frequency over the temperature range (−50 to 150 °C) for all of the fibers. The storage modulus of the PVDF/rGO nanocomposite fibers had a higher value (7.5 GPa) in comparison with other fibers. The creep and creep recovery behavior of the PVDF/nanofillers in the nanocomposite fibers have been explored in the linear viscoelastic region at three different temperatures (10–130 °C). In the PVDF/rGO nanocomposite fibers, strong sheet/matrix interfacial interaction restricted the mobility of the polymer chains, which led to a higher modulus at temperatures 60 and 130 °C.

## 1. Introduction

Due to their unique ability in energy conversion from ambient vibration energy to electric potential, piezoelectric materials have received great interest. Piezoelectric materials are considered in applications including energy harvesting, actuators, sensors, and health monitoring devices due to their specific energy conversion [1,2]. However, time-dependent creep deformation can affect the implementation and consistency of piezoelectric materials. Non-stationary creep analysis of these materials, which provides their history of stresses and deformations, is essential for the reliability of these components [3]. The creep deformation of polymers is strongly dependent on the mobility and orientation of the polymer chain as well as the polymer molecular weight.

One of the major driving forces for structural failure in practical applications is creep-induced plastic deformation [4]. The study of mechanical creep and relaxation of polyvinylidene fluoride (PVDF) has appeared to be of significant parameters for use in microphones, sensors, and transducer [5].

PVDF is a semi-crystalline polymer that has been widely considered for its piezoelectric properties, good chemical resistance, strength, and thermal resistance [6]. PVDF has four different crystallized forms that involve three different chain conformations, such as the α, β, γ, and δ phases. Among them, the β-phase has the largest spontaneous polarization per unit cell and therefore presents the highest electro active properties [7]. The β-content in PVDF was found to increase by about 8% with a silver nanoparticle (Ag-NPs) content of 0.4%, which led to a good piezoelectric response [8]. The proportion of the β-phase (F(β)) to the total crystalline phase for PVDF/SiO_2_ (2 wt.%) increased when compared to that of neat PVDF [9].

Hence, many works have been dedicated to the development of PVDF β-phase formation. The reported methods to enhance β-phase in PVDF are filler addition, blending, poling, stretching, or nanoconfinement [10]. Although these techniques have high efficiency in β-phase formation, they suffer from challenges associated with homogeneous filler dispersion (composite preparation), phase separation (polymer blending), or exclusive processes that have a long duration (poling, stretching, and nanoconfinement), which limit their industrial processing phase [11]. Generally, thermoplastics have been recognized to demonstrate a complex non-linear response to externally applied loads. Due to their low friction, fluorinated thermoplastics are even more complicated. The mechanical response is comprised of linear viscoelastic behavior at small strains followed by a distributed yielding that progresses to a large-scale viscoplastic flow as the strain increases until gradual material stiffening develops prior to ultimate failure [12]. Research has indicated that adding nanofillers into the polymers in the composite structure is an effective approach to enhance the mechanical behavior of polymers [13].

The addition of nanoparticles to piezoelectric polymers is well established as a means to improve performance. The electrostatic interactions between the positive and negative charge centres of metal ions and negative charge clouds in rGO leads to enhancement in the PVDF piezoelectric polar phases. Reduced graphene oxide stabilizes the polar phase in the PVDF and easily leads the charges toward the electrodes and enhances the energy harvesting properties [14]. The generated output voltage was 184 mV under an applied force of 2.125 N with the piezoelectric sensitivity of 173.507 mV/Nμm for PVDF/PZT nanocomposite fibers with 0.37 PZT volume fractions [15].

Energy harvesting from hybrid piezoelectric composites based on barium titanate (BT) is desirable due to its high dielectric constant (ε ∼ 2000), low cost, natural abundance, and ecofriendly properties. Although BT has the limitation of flexibility and durability, it still remains an unavoidable challenge in enhancing the performance of piezo generators [16].

The integration 1.93 wt% of a multi wall carbon nanotube into PVDF improved its storage modulus by 100–150% over a wide range of temperatures [17]. The addition of carbon nanotubes into PVDF considerably improved its creep resistance [13].

Due to the wide range of applications of PVDF composites, they will be subjected to huge thermal gradients over time, crossing the glass transition temperature of the material. Therefore, considerable changes may be introduced within the polymeric microstructure leading to relevant alterations in its macroscopic behaviour [18].

Our previous research showed that novel triaxial braided PVDF yarn generated a maximum output voltage of 380 mV and a power density of 29.62 μW cm^−3^, which is ∼1559% higher than previously reported for piezoelectric textiles [19]. Additionally, it was found that high performance PVDF/BT hybrid piezofiber can charge a 10 μF capacitor in 20 s, which is four and six times faster than previously reported for PVDF/BT and PVDF energy generators [16].

Adding BT as the piezoceramic filler enhances charge generation properties in the matrix due to the delivery of low breakdown strength and a high dielectric constant [20]. On the other hand, the electrostatic interactions between negative charge clouds in reduced graphene oxide and the positive and negative charge centers of metal ions improve the polar phase in the piezoelectric material. This stabilized polar phase in PVDF/rGO composite easily leads charge movement toward electrodes, boosting energy harvesting properties [21]. The idea of mixing these two fillers in the PVDF matrix was to take advantage of both these two fillers. The PVDF/rGO/BT coil fiber generated a voltage output of ≈1.3 V and a power density of 3 W Kg^−1^ under ≈100% strain with the energy conversion efficiency of 22.5% [22].

The melt-spun piezoelectric fibers of PVDF, PVDF/BT, PVDF/rGO, and PVDF/rGO/BT were fabricated and discussed in our previous work, and the effects of each these fillers on piezoelectric performance were discussed [16,19,22]. In continuation of our previous work, due to the wide range of applications of these high-performance nanocomposite fibers, in this work, the influences of various combinations of PVDF and nanofillers have been investigated on storage modulus and creep behavior.

## 2. Materials and Methods 

### 2.1. Materials

Polyvinylidene fluoride (PVDF) powder was purchased from Solvay Soleris (Milan, Italy) under the commercial name Solef 6010. The melt flow index (MFI) of Solef 6010 is 2 g/10 min at a load of 2.16 kg (or 6 g/10 min at a load of 5 kg) at 230 °C. The cubic barium titanate nanoparticles with a mean diameter of 50 nm and with a 99.9% trace metal base were purchased from Sigma Aldrich Company (Beijing, China). *N*,*N*-dimethylformamide (DMF, >99.8%, Merck, Kenliworth, NJ, USA) was used as a solvent.

### 2.2. Methods

#### 2.2.1. Preparation of the Nanocomposites

Reduced graphene oxide in a concentration of 1 mg mL^−1^ was dispersed in DMF according to the method we reported previously [23,24]. Briefly, using hydrazine and ammonia at 90 °C, an exfoliated aqueous graphene oxide (GO) dispersion (0.05 wt.%) was chemically reduced in two steps. The addition of H_2_SO_4_ (5 wt.%) to the dispersion followed by filtration and drying resulted in the formation of agglomerated graphene powder. The graphene powder was then dispersed in DMF using triethylamine through sonication and centrifugation cycles to prepare a stable dispersion.

To prepare the PVDF/rGO composite, 0.5 wt.% rGO, was added to a PVDF/DMF (15 *w*/*v*%) under constant stirring and sonication for 2 h to reach the homogeneous dispersion of the rGO nanosheets in the polymer matrix. After film casting, the film was left until it was dry and all of solvent had evaporated. The composite film was chopped into small pieces, washed in ethanol, and dried in a vacuum oven at 50 °C for 4 h.

To prepare the PVDF/BT nanocomposite, 10 wt% of BT nanoparticle was dispersed into DMF (50 mL) using a probe sonicator for 60 min under nitrogen flow at 0 °C. The prepared solution was added to the PVDF/DMF solution and was sonicated and stirred for 1 h. To prepare the PVDF/rGO/BT nanocomposite, the as-prepared PVDF/rGO/DMF solution was mixed with dispersed BT/DMF and was sonicated for 30 min under nitrogen flow followed by stirring for 2 h. The suspensions were cast onto a glass plate to evaporate the solvent until the film could be peeled off from the dish. The composite film was chopped into small pieces. The chopped composite films were heated overnight at a temperature 70 °C. More details on the PVDF nanocomposite fabrication procedure are described in our previous works [16,19,22].

#### 2.2.2. Melt Spinning of Nanocomposite Fibers

Melt spun fiber was fabricated by a twin screw extruder (Barrel Scientific Ltd., Melbourne, Australia) with a single hole spinneret with a 2 mm diameter. The chopped composite films were fed directly into the extruder. To fabricate a uniform fiber diameter, the volume of the nanocomposites flowing through the spinneret and that were taken up were speed were controlled. The temperature profiles along the extruder varied through the nine sequential zones ranging from 180 to 220 °C. Figure 1 shows the procedure used for fabricating the nanocomposite fibers. More information is available in our recently published papers [16,19].

#### 2.2.3. Characterization

DMA tests were performed using DMA 242 Artemis from NETZSCH (Selb, Germany). The storage modulus (E′), loss modulus (E″), and loss factor (tan δ = E″/E′) of PVDF and its nanocomposite fibers were measured in tensile mode following a heating ramp at 3 °C min^−1^ from −50 °C to 150 °C in various frequencies (0.1, 0.2, 0.5, 1, 2, 5, 10 and 20 Hz). The amplitude of the dynamic stress was set at 0.2 MPa, well within the linear viscoelastic range of the samples. All of the measurements were repeated five times. The melting temperature (T_m_) and melting enthalpy (ΔH_m_) of the fibers were evaluated with a differential scanning calorimeter Phoenix from NETZSCH, Germany, at a heating rate of 10 °C min^−1^. The surface morphology of the as-prepared nanocomposite fibers were analyzed with the use of a JEOL 7500 SEM (JEOL Ltd., Tokyo, Japan). Secondary electron imaging was used at an accelerating voltage of 10 kV and a probe current setting of 30 nA.

## 3. Results and Discussion

### 3.1. Crystalline Structure

The dispersion of filler in the matrix and the interaction between the matrix and filler are critical for enhancing the different properties of the PVDF matrix [25]. Numerous studies have discovered that to improve the physical and thermal behaviors of a composite, its morphology is a considerable parameter [26]. The surface morphological features of the PVDF nanocomposite fibers were characterized using SEM, as shown in Figure 2. The dispersion of the fillers in the PVDF matrix were evaluated by EDS, SEM, and XRD analysis from the fiber length and the cross section. Additionally, the crystalline structures presented here from the neat PVDF and its composites fibers obtained from the optimized concentration of fillers, as previously established and reported upon in our previous works. The amount of 0.5 wt% rGO as a conductive filler and 10 wt% BT as piezoceramic fillers were chosen based on the best piezoelectric performance that these nanocomposite fibers presented in comparison with other concentrations. [11,14,17,27].

The optimized concentration avoids any clustering of the added particles. When stress is applied, these clusters may initiate failure processes, which leads to a reduction in the strength of the composite [28]. The surface morphology and cross-section of the PVDF and hybrid PVDF/BT fibers show that they are very smooth and do not have any porosity or voids (Figure 2a–d). The homogeneous dispersion of the filler throughout the nanocomposite can be clearly observed in Figure 2e,f from the cross section of the PVDF/rGO fiber. Figure 2f reveals the low content of graphene nanosheets with the dimension of 200–400 nm. The excellent dispersion of rGO in DMF would finally lead to its excellent dispersion in the PVDF matrix. Figure 2g,h shows the surface and cross section of the PVDF/rGO/BT fiber with nanofillers dispersed uniformly in the PVDF with no visible aggregation, indicating good compatibility between the polymer and the nanoparticles [29]. It was shown that interaction between the BT and rGO nanostructures with PVDF chains leads to β phase formation. The BT nanoparticles and rGO can attract PVDF chains, which crystallize on their surfaces in the all-transform due to the interfacial interactions, facilitating the transformation of the local amorphous phase and the α phase into the β phase. In this case, the BT nanoparticles and rGO act as nucleating agents, providing the substrates for the formation of PVDF crystalline nucleation and inducing the formation of the β phase of the PVDF segment via the strong interactions at the interfaces [22].

In our previous work [22], we observed better mechanical properties in PVDF fibers containing rGO, which can be attributed to the uniform dispersion of graphene sheets into the polymer matrix as well as the high intrinsic strength of graphene. The final diameter of the stretched PVDF and PVDF nanocomposite fibers was ≈170 µm.

### 3.2. Differential Scanning Calorimetry (DSC)

The effect of the nanofiller addition on the crystallization and melting temperatures of the nanocomposites were determined by DSC analysis. The second heating and cooling characteristics of the samples at a scanning rate of 5 °C/min were monitored, and the curves that were obtained are depicted in Figure 3. The nanoparticle effects on the thermal properties of PVDF/BT are clearly seen from the peak shifts. The melting temperature (T_m_) and crystallization temperature (T_c_) slightly increase in the nanocomposite with the addition of the filler (Table 1). This increment is a result of the homogeneous dispersion of the filler throughout the matrix, which has the role of a nucleating agent and prevents segmental movement of the polymer chain [30]. The crystallinity percentage (χc) was determined from the following Equation (1):(1)$$X_{c}=\frac{\Delta H_{m}}{\left(1-\varphi \right)\Delta H_{m}^{0}}\times 100$$
where ΔH_m_ is the nanocomposite melting enthalpy, $\Delta H_{m}^{0}$ is the melting enthalpy of the 100% crystalline PVDF (103.4 Jg^−1^), and φ is the weight percentage of the nanocomposite filler [31].

The pure PVDF showed a melting peak at 172.4 °C, which was slightly lower than that for PVDF/rGO nanocomposites at 172.6 °C. The PVDF β phase has a higher melting temperature than the α phase due to has trans conformation, which allows the chains to be packed more compactly in crystal structure [31,32]. As a result, the predominant phases for the pure PVDF and PVDF/rGO nanocomposites were mainly the α phase and the β phase. This conclusion was confirmed by the FTIR and XRD results [22]. The cooling DSC curve indicated that crystallization happened at a slightly higher temperature (T_c_) for nanocomposite fibers in comparison with pure PVDF fibers.

The crystallization temperature increased for the composites made with rGO due to rGO’s very large surface area, which adsorbs PVDF chains and causes easier nucleation [31]. The PVDF/rGO/BT and PVDF/rGO nanocomposites exhibited similar crystallization behaviors. The difference between them is that the PVDF/rGO nanocomposite has a higher T_c_ (143.3 °C) compared to the PVDF/rGO/BT nanocomposite (143 °C). A possible reason for this could be a change in the dispersion quality of the BT particles after adding the rGO nanosheet. Many of the BT particles are deposited on the rGO sheet, consequently reducing PVDF crystallite growth [33]. The crystallinity temperature variation for nanocomposites depends on the dimensions, filler concentration, and the interfacial interactions [30].

The crystallinity percentage (X_c_) was determined from Equation 1:(2)$$X_{c}=\frac{\Delta H_{m}}{\left(1-\varphi \right)\Delta H_{m}^{0}}\times 100$$
where ΔH_m_ is the nanocomposite melting enthalpy, $\Delta H_{m}^{0}$ is the melting enthalpy of the 100% crystalline PVDF (103.4 Jg^−1^), and φ is the weight percentage of the nanocomposite filler [31]. In all of the samples, the overall degree of crystallinity changed only slightly—from 41 to 43% (Table 1).

### 3.3. Dynamic Mechanical Behaviour

DMA is a useful technique for the experimental characterization of the small-strain viscoelastic properties of polymers. During the dynamic deformation of materials, the storage modulus (E′) represents its elastic behavior, which is responsible for its time-independent recovery, and the loss modulus (E″) represents the viscous behavior, which is responsible for its irreversible energy dissipation. The ratio of these two, defined as tan δ = E″/E′, where the phase transition temperatures can be identified by locating the position of the tan (δ) peaks via a temperature-sweep experiment [34].

Generally, the incorporation of rigid nanofillers into the polymer matrix due to the strong interfacial interaction or the bond formation between the polymer matrix and the nanofiller results in an effective load transfer from the polymer matrix to the nanofiller, thereby increasing the mechanical properties [35]. For PVDF composite fibers, the high density of the polar interactions between the fillers and the polymer molecules, such as electrostatic action and hydrogen bonding, are often considered to favor the formation of the β-phase, which provides more crystallinity [36].

The polymer chain motions are classified depending on their nature in the α, β and γ transitions. At low temperatures, the local motions, such as the rotation of the lateral groups (γ relaxation) or the motions of the main chain segment (β relaxation), lead to secondary relaxations. At higher temperatures, the α relaxation relates to the cooperative motions of the main chains [37]. Note that these molecular relaxations are not associated with the PVDF crystalline phases that are also denoted α, β, γ, and δ by convention.

The low value of the loss factor means that the polymer composites have a more ideal, solid-like behavior [38]. For the determination of the dynamic mechanical properties of the PVDF and its nanocomposite fibers, the storage modulus (E′) and loss factor (tan δ) of all of the samples were analyzed as functions of temperature in different oscillation frequencies. As it can be observed, with an increase in the frequency, the storage modulus increased for all of the samples due to good particle dispersion. It can be inferred that the inclusion of nanofiller in the polymer matrix increases the storage modulus to a definite point [39]. The results demonstrated that the inclusion of BTO in the PVDF blend matrix resulted in a notable change in the stiffness of the composite materials, which indicates an excellent reinforcing effect of the BTO.

Additionally, E′ decreases based on temperature in all of the samples and at all of the frequencies due to the restricted motions of the main chains (Figure 4).

The T_g_ is extremely dependent on the frequency due to the viscoelastic behavior of polymers. The T_g_ values of samples were estimated from peaks in the tan δ versus the temperature curves shown in Figure 5. An increase in frequency means a a decrease in the polymer chain relaxation time [40]. Since the relaxation time is viable at higher temperatures, the chain’s preference is to achieve the relaxation time at high temperatures. A comparison between the T_g_ values (the peak of the curve) at a constant frequency reveals that the T_g_ of the neat polymer is lower than its nanocomposite. At higher temperatures, the nanoparticles reduced the free volume between the chains, blocking their movements, as a result their relaxation time increase [29].

A comparison of the neat and composite samples is given in Figure 6. In Figure 6a, the storage modulus for pure PVDF fibers was found around 3.5 GPa at −50 °C and decreased over the temperature range −50 to 150 °C. PVDF/BT, PVDF/rGO/BT, and PVDF/rGO show temperature dependences with a trend that is similar to that of PVDF. Moreover, it was discovered that the storage modulus of the nanocomposite fibers is higher than pure PVDF, with PVDF/rGO having the highest E′ compared to the other fibers. This can be attributed to the reinforcing effect [41] of rGO. rGO sheets have a high aspect ratio with strong interfacial adhesion for the integration in the PVDF matrix [42]. The good miscibility and fine dispersion of rGO sheets in the PVDF matrix, as confirmed from the SEM images, is the other important contributing factor [43]. By increasing the temperature to above 25 °C, the E′ decreased to values smaller than that of PVDF in all of the nanocomposite fibers. A decrease in the moduli of the nanocomposite fibers could be due to the filler aggregation in the matrix of the PVDF, which reduces the effective interfacial area between the filler and the polymer chains [4].

Figure 6b shows two relaxation peaks at low and high temperatures for PVDF based fibers. PVDF exhibits peaks at −44.9, and 85.5 °C, corresponding to the glass transition temperature (T_g_) and the relaxational transition temperature (T_r_), respectively. The T_g_ results from the start of the segmental motion of PVDF’s amorphous portion. The T_g_ increases by increasing the filler loading because of the strong cohesive force between the filler, which makes the polymer more compact, and decreasing the free space for polymer segmental motion. The increase in tan (δ) above −50 °C is according to the PVDF dipolar behaviour temperature as well as BT. By increasing the temperature above −50 °C, the properties are mostly affected by dipolar polarization. There is a phase transformation above −70 °C for BT particles from rhombohedral to orthorhombic. Consequently, an increase in tan (δ) in this temperature region (−70–3 °C) is predictable [44]. The Tr indicates the relaxation of the PVDF chains at the crystalline region where the segments are in a limited state [45]. The position of the Tr peak for the pure PVDF and the PVDF nanocomposites is different. The increase of difference in transition temperatures for relaxation in the crystalline region could be ascribed to the PVDF polymorphic structure variation from the α to β phase. The β phase has denser volume (density = 1.99 g cm^−3^) compared to the α phase (density = 1.92 g cm^−3^) and the amorphous phase (1.78 g cm^−3^); hence, it needs more thermal energy for the relaxation to take place. A sharp peak in the relaxation temperature occurs for the PVDF nanocomposites when the β phase conversion is complete [46]. By adding fillers, the PVDF crystalline phase shifts from the α to β phase due to the presence of a more packed chain structure. Additionally, compared to α-PVDF, β-PVDF shows a more packed crystalline structure [11].

### 3.4. Tensile Creep Behaviour

Material deformation occurs under constant stress during time known as creep. At temperatures above T_g_, most amorphous polymers are recognized as thermo-rheologically [47]. In polymers, the deformation evolution under a constant and continuous load is highly dependent on the chain’s mobility. The mechanism of uncoiling polymer chains, the untwisting, unravelling, untangling, or slippage of the polymer molecules past one another, depends on the molecular structure and nature of the stress applied [48]. The creep behaviour for semicrystalline polymers becomes complicated due to the complex coupling between the crystalline and amorphous phases. By keeping temperatures above T_g_, if the creep strain is limited to below the yield strain, the creep behaviour is mostly controlled by the amorphous phase deformation in semicrystalline polymers [13].

PVDF creep behaviour and its nanocomposite filaments along the longitudinal direction were investigated at 10, 60, and 130 °C (Figure 7). Creep behavior usually includes three stages: elastic deformation, primary, and secondary creep. It can be seen that the creep and recovery response of the PVDF nanocomposite fibers increases (strain) with increasing temperature.

At low temperatures, low instantaneous deformation is observed for the PVDF/BT fibers due to the smaller number of activated polymer segments that need to be involved in the creep response. By increasing the temperature, more segments are activated and introduce the relative velocity of the orientation of the polymer segments and entanglements; therefore, a higher creep rate occurs (Figure 7a–c). Pure PVDF fiber tends to become softer with temperature, which leads to higher creep at 60 and 130 °C in comparison with PVDF/rGO and PVDF/rGO/BT.

PVDF/rGO fibers exhibit lower creep strain at 60 and 130 °C, indicating their higher strength compared to the other samples. When incorporated into the polymer matrix, the graphene sheets form a three-dimensional network with the polymer chains, confining their movement and disentanglement under stress. This results in a great enhancement in the polymer stiffness. The combination of rGO and BT has apparently resulted in a crowding effect, as evident by a strain value close to pure PVDF. Poor interaction between the fillers can create filler aggregations, which significantly decreases the tensile strength of a sample [48].

The relaxation process in the composite structure is a reflection of the resistance of the matrix in rearranging the polymer segments between the crosslinks due to some internal/external imposed stress [49]. The recovery analyses at different temperatures are presented in Figure 7d–f. The recovery responses of the fibers at all temperatures show that the PVDF/rGO fiber has a very high recovery. This may be due to the degree of crystallinity of the PVDF/rGO fiber. The effect of the temperature increasing from 60 to 130  °C on the recovery behavior of all of the fibers is presented in Figure 7e,f, respectively. At these temperatures, which are very close to the softening point of a polymer, the crystalline regions become unstable, and therefore, their role in increasing recovery diminishes. In this situation, rGO has a strong effect on improving the creep resistance of PVDF polymer, which may be the reason for the high strain recovery in PVDF/rGO. In general, the addition of rGO was found to have an enhancing effect on the strength and modulus of PVDF. The interaction between graphene and the host polymer consists of Van der Waals interactions, physical bonding, and mechanical interlocking [50]. The functional rGO groups also interact with the polymer chains and form an interphase zone around the rGO nanosheets, which can significantly improve the elastic properties of the matrix [51]. When the composite is subjected to a force, the load is be transferred from the polymer chains to the graphene sheets to decrease the stress concentration in the matrix, which can result in an increase in the mechanical properties of the nanocomposite.

## 4. Conclusions

Nanostructured piezoelectric PVDF nanocomposite fibers were developed through the melt-spinning process for applications in generators and sensors in wearable technology. The DSC crystallization findings revealed improvement in the crystallization of PVDF nanocomposite fibers. DMA results showed that storage modulus of the samples improved in the studied temperature range. T_c_ increased by approximately 3–4 °C, which approved chain mobility restriction in the nanocomposite. The tan δ peak curves displayed two peaks related to the T_g_ of the polymer and the release of a constrained phase due to adding fillers. The relaxation temperature had a sharp peak for the PVDF nanocomposites due to β phase formation. For semicrystalline polymers, the creep behaviour becomes complicated due to the complex coupling between the crystalline phase and the amorphous phase. The study demonstrated that the creep resistance of PVDF can be improved significantly upon addition of a rGO and rGO/BT mixture. The dynamic mechanical properties of PVDF and its nanocomposites with rGO and BT were studied in this work and showed the potential to predict the long-term creep performance of these materials.

## Figures and Tables

**Figure 1 nanomaterials-11-02153-f001:**
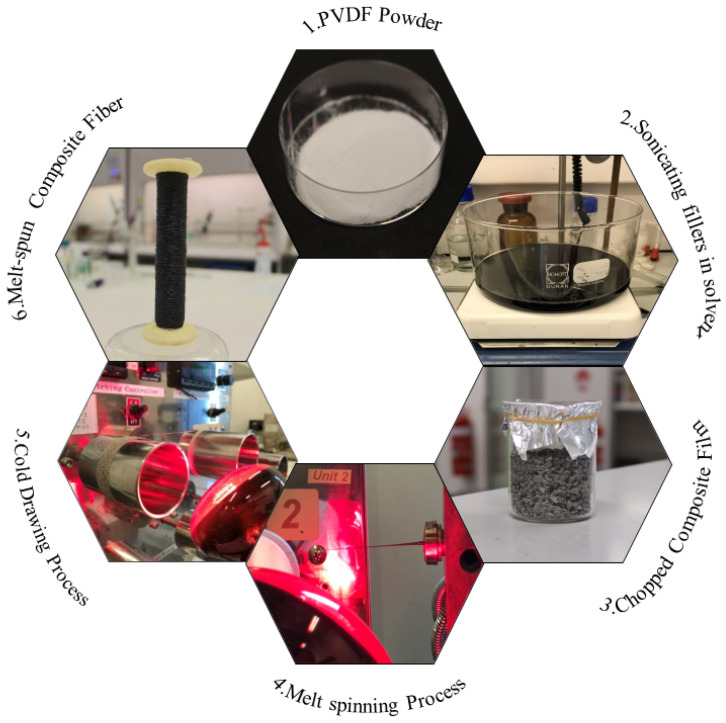
Preparation process of PVDF/rGO nanocomposite fibers.

**Figure 2 nanomaterials-11-02153-f002:**
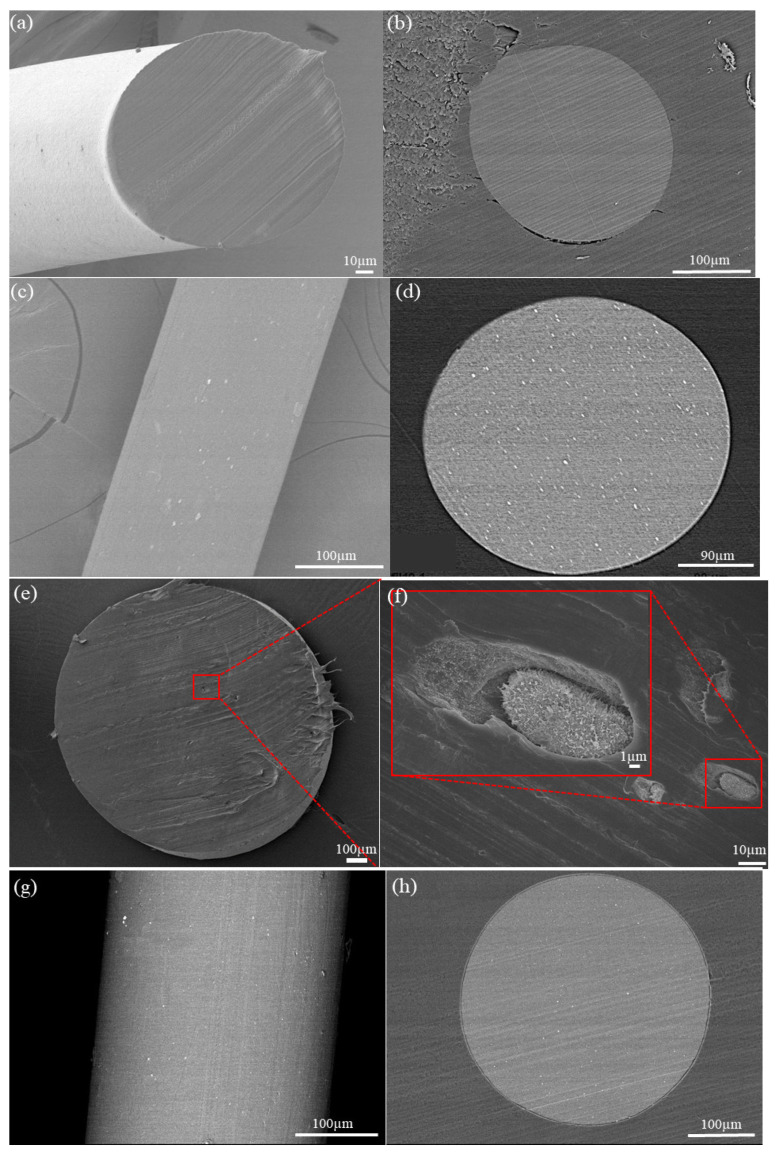
SEM images of the nanocomposite fibers:(**a**) surface and (**b**) cross section of PVDF fiber; (**c**) surface and (**d**) cross section of PVDF/BT (10 wt%); (**e**,**f**) low and higher magnification of cross-section of PVDF/rGO fiber (0.5 wt%) with the low content of graphene nanosheets with the dimension of 200–400 nm; and (**g**,**h**) surface and cross section of PVDF/rGO/BT fiber.

**Figure 3 nanomaterials-11-02153-f003:**
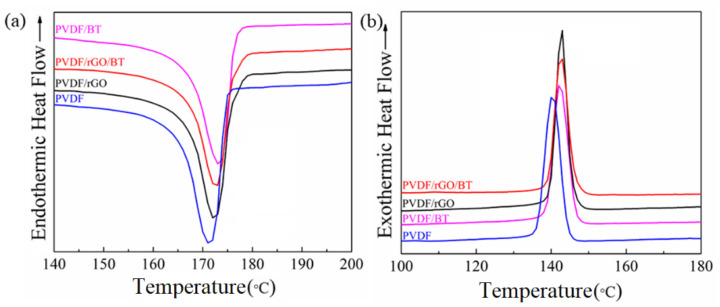
DSC second melting (**a**) and cooling (**b**) curve of PVDF, PVDF/BT, PVDF/rGO, and PVDF/rGO/BT nanocomposite fibers.

**Figure 4 nanomaterials-11-02153-f004:**
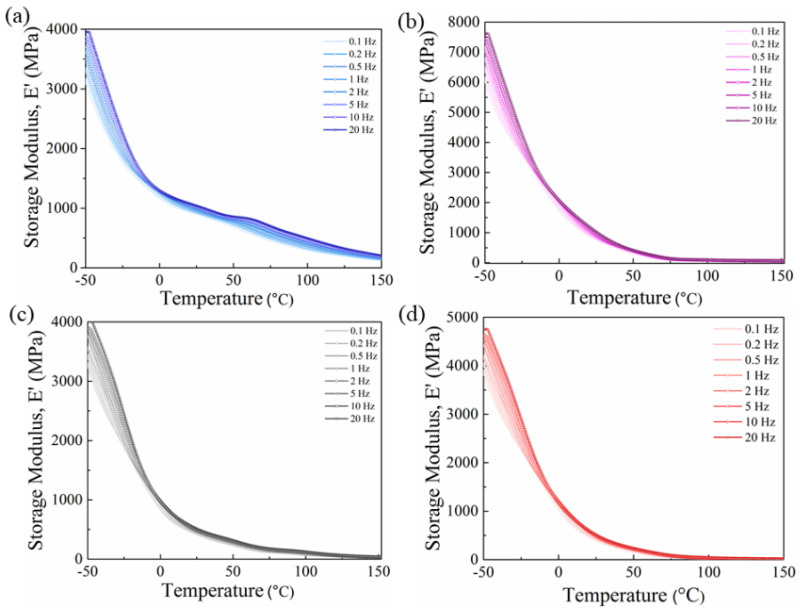
Storage modulus vs. temperature for (**a**) PVDF, (**b**) PVDF/BT, (**c**) PVDF/rGO, and (**d**) PVDF/rGO/BT fibers obtained from DMA measurement. In all cases, the storage modulus was determined at a range of oscillation frequencies, as indicated.

**Figure 5 nanomaterials-11-02153-f005:**
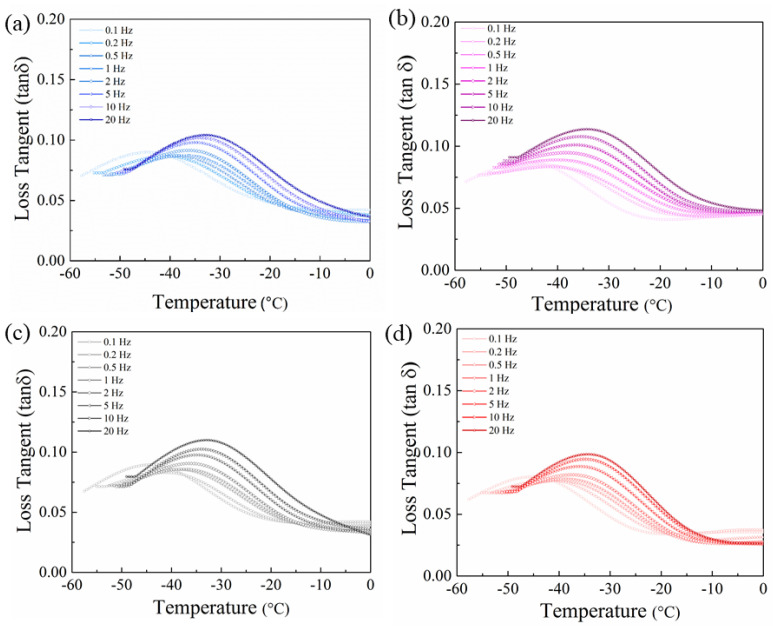
Temperature and frequency dependence of tan δ for (**a**) PVDF, (**b**) PVDF/BT, (**c**) PVDF/rGO, and (**d**) PVDF/rGO/BT nanofibers.

**Figure 6 nanomaterials-11-02153-f006:**
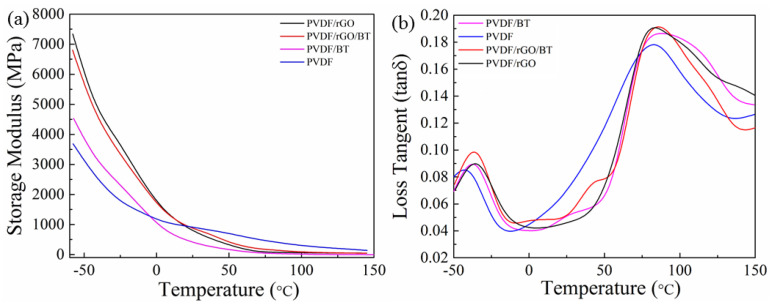
Dynamic mechanical analysis of the PVDF nanocomposite fibers at frequency 1 Hz: (**a**) storage modulus, (**b**) loss tangent (tan δ).

**Figure 7 nanomaterials-11-02153-f007:**
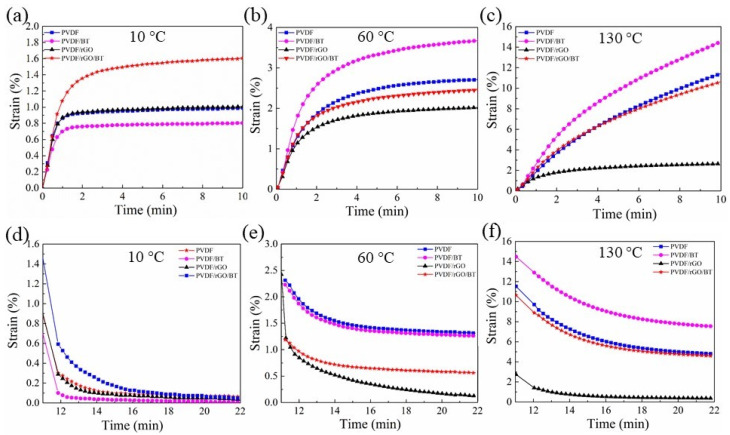
The creep behaviours (**a**–**c**) and creep recovery (**d**–**f**) of the melt-spun PVDF, PVDF/BT, PVDF/rGO, and PVDF/rGO/BT fibers at temperature (10, 60, 130 °C).

**Table 1 nanomaterials-11-02153-t001:** DSC results for PVDF and its nanocomposite fibers.

Samples	T_m_ (°C)	T_c_ (°C)	X_c_ (%)
PVDF	172.4	140.1	41
PVDF/BT	173.3	142.4	42
PVDF/rGO	172.6	143.3	43
PVDF/rGO/BT	172.5	143	43

## Data Availability

Not applicable.

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
