# Peer review of "Dynamic Mechanical and Creep Behaviour of Meltspun PVDF Nanocomposite Fibers"

_nanomaterials, 2021, doi:10.3390/nano11082153_

Round 1

Reviewer 1 Report

Manuscript ID: nanomaterials-1328132

Title: Dynamic mechanical and creep behaviour of meltspun PVDF nanocomposite fibers

In the present manuscript, dynamic mechanical and creep behaviour of 2 meltspun PVDF nanocomposite fibers were investigated. The manuscript in the present form is well-organized. However, some major revisions will help the authors to improve the quality of the paper. Below, my comments are listed.

  • Some typos and grammatical errors are in the text. The text should be precisely rechecked. I.e. “…..creep and 26 creep recovery behavior of PVDF/nanofillers in the nanocomposites fibers has (have) been explored…”.
  • Some characterization regarding the BT nanoparticles and rGO such as FTIR, FE-SEM, TEM, so on should be added.
  • What is the interactions between BT and rGO nanostructures with PVDF chains? Their interaction will help author to interpretate better the changing in Betha-phase contents due to the interactions of PVDF chains with functional groups of nanostructures.
  • The author should discuss how nanoparticles change the piezoelectric behavior of PVDF? Use the literature and a schematic illustration is suggested.
  • rGO nanosheets are black but the nanocomposite shown in Figure 1 are white. Please clarify.
  • The contents of rGO and BT in the composite fibers are 0.5 and 10 wt.%, respectively. How do the authors compare the results of BT/PVDF with rGO/PVDF?
  • The overall morphology of the melt-spun PVDF fibers must be assessed by FE-SEM and fiber diameter distributions of different systems should be added in the manuscript. The author should discuss the effects of each nanostructure on the decrement/increment of the fibers’ diameters.
  • How did the author understand the proper dispersion of nanostructures in the fibers? Some techniques should be done on the fibers to present the uniform dispersion.
  • rGO nanostructures are partially conductive based on the content od the structural defects. How the increment of the conductivity of PVDF fibers change the piezoelectric properties of the final composite fibers?
  • The author can use the following papers to improve the quality of their paper.

Chamankar, N., Khajavi, R., Yousefi, A. A., Rashidi, A., & Golestanifard, F. (2020). A flexible piezoelectric pressure sensor based on PVDF nanocomposite fibers doped with PZT particles for energy harvesting applications. Ceramics International, 46(12), 19669-19681.

Haddadi, Seyyed Arash, Ahmad Ramazani SA, Soroush Talebi, Seyyedfaridoddin Fattahpour, and Masoud Hasany. "Investigation of the effect of nanosilica on rheological, thermal, mechanical, structural, and piezoelectric properties of poly (vinylidene fluoride) nanofibers fabricated using an electrospinning technique." Industrial & Engineering Chemistry Research 56, no. 44 (2017): 12596-12607.

Issa, A. A., Al-Maadeed, M. A., Luyt, A. S., Ponnamma, D., & Hassan, M. K. (2017). Physico-mechanical, dielectric, and piezoelectric properties of PVDF electrospun mats containing silver nanoparticles. C, 3(4), 30.

Hosseini, S. M., & Yousefi, A. A. (2017). Piezoelectric sensor based on electrospun PVDF-MWCNT-Cloisite 30B hybrid nanocomposites. Organic Electronics, 50, 121-129.

Author Response

Dear Editors and Reviewers:

We would like to first thank the editor and reviewers for their feedback. The reviewers’ comments are specifically addressed below, with reference to the point in the manuscript into which feedback has been incorporated. Responds to the reviewer’s comments:

Reviewer #1:

In the present manuscript, dynamic mechanical and creep behaviour of 2 meltspun PVDF nanocomposite fibers were investigated. The manuscript in the present form is well-organized. However, some major revisions will help the authors to improve the quality of the paper. Below, my comments are listed.

  1. Some typos and grammatical errors are in the text. The text should be precisely rechecked. I.e. “…..creep and 26 creep recovery behavior of PVDF/nanofillers in the nanocomposites fibers has (have) been explored…”.
    - The manuscript text was rechecked.

  2. Some characterization regarding the BT nanoparticles and rGO such as FTIR, FE-SEM, TEM, so on should be added.
    - The aim of this paper was just DMA analysis of nanocomposite fiber with high piezoelectric performance that we presented in our previous papers. The more details about each of these nanocomposite fibers presented in our previous paper (references 10,16,19,22) as mentioned in the manuscript as well.

  3. What is the interactions between BT and rGO nanostructures with PVDF chains? Their interaction will help author to interpretate better the changing in Betha-phase contents due to the interactions of PVDF chains with functional groups of nanostructures.
    - The following description added to the manuscript:
    “The BT nanoparticles and rGO can attract PVDF chains to crystallize on their surfaces in the all-trans form due to the interfacial interactions, facilitating the transformation of local amorphous phase and α phase into the β phase. In this case, the BT nanoparticles and rGO act as nucleating agents, providing the substrates for the formation of PVDF crystalline nucleation and inducing the formation of β phase of PVDF segment via the strong interactions at the interfaces.”

  4. The author should discuss how nanoparticles change the piezoelectric behavior of PVDF? Use the literature and a schematic illustration is suggested.
    - The aim of this manuscript is dynamic mechanical analysis of PVDF nanocomposite fiber. The details of piezoelectric behaviour of each nanocomposite fibers were presented in our previous works as mentioned in the references 10,16,19,22.

  5. rGO nanosheets are black but the nanocomposite shown in Figure 1 are white. Please clarify.
    - Figure 1 was updated.

  6. The contents of rGO and BT in the composite fibers are 0.5 and 10 wt.%, respectively. How do the authors compare the results of BT/PVDF with rGO/PVDF?
    -As mentioned in line 174 the fillers concentration optimization was done and explained in detail in our previous works (reference:10,16,19,22). The amount of 0.5 wt% rGO as conductive filler and 10 wt% BT as piezoceramic fillers were chose based on best piezoelectric performance that these nanocomposite fibers presented in compare with other concentration.

  7. The overall morphology of the melt-spun PVDF fibers must be assessed by FE-SEM and fiber diameter distributions of different systems should be added in the manuscript. The author should discuss the effects of each nanostructure on the decrement/increment of the fibers’ diameters.
    -All PVDF and its nanocomposite fibers fabricated by melt spinning process which through size of die, feeding rate, and collector tension the fibers diameter are controllable. In all these composite fibers to just evaluate the effect of filler tried to keep fiber diameter constant. As mentioned in the text line 196 “The final diameter of the stretched PVDF and PVDF nanocomposite fibers was ≈170 μm.’’

  8. How did the author understand the proper dispersion of nanostructures in the fibers? Some techniques should be done on the fibers to present the uniform dispersion.
    - The dispersion of fillers in PVDF matrix evaluated by EDS, SEM, and XRD analysis from fiber length and cross section in our previous works which also mentioned and referred to them in this manuscript. Since we discussed about all these details in our previous works, we just referred to them in this paper (reference:10,16,19,22) and focus on the aim of this work which is dynamic mechanical analysis of composite fibres.

  9. rGO nanostructures are partially conductive based on the content of the structural defects. How the increment of the conductivity of PVDF fibers change the piezoelectric properties of the final composite fibers?
    - Although the aim of this work in dynamic mechanical analysis and not about piezoelectric properties but the following description added to the text:
    “The electrostatic interactions between the positive and negative charge centres of metal ions and negative charge clouds in rGO leads to enhancement in the PVDF piezoelectric polar phases. Reduced graphene oxide stabilizes polar phase in PVDF and leads the charges easily toward the electrodes and enhances the energy harvesting properties.”

  10. The author can use the following papers to improve the quality of their paper.
    - Thanks for suggestion, referred to all these valuable papers in the manuscript (reference: 8,9,14,15).

Reviewer 2 Report

In their paper "Dynamic mechanical and creep behaviour ...", Foroughi and coworkers give an overview over some important mechanical properties of certain nanocomposite piezoelectric fibers. Even so these properties are explained in a rather qualitative and descriptive way, their knowledge is for sure of major importance for the development and characterization of nanocomposite fiber based devices and thus of interest for the readers of nanomaterials. The methodology sounds solid, and the publication continues previous work by the same authors in the field.

However, some questions remain.

  1. line 73: the abbreviation "MWNT" has not been defined before.

  2. line 144: both the storage modulus as well as the loss modulus are called E'.

  3. line 152: shouldn't the unit of "probe current" be mA instead of mV?

  4. Figure 2: The authors should provide some more details on what exactly is shown in Figure (f).

  5. Figure 3: in both figures (a) and (b) the x-axis label should be "Temperature (°C)", not "Tempreture (°C)".

  6. Table 1: what are the error bars for Tm, Tc, and Xc? The same for the peak temperatures in line 307.

  7. Figure 7: Shouldn't (f) also be "creep recovery"? In the caption only d and e are mentioned.

  8. The authors should check the references. In line 508, a "n/a" is given for year or volume. For ref 41 (line 524) the journal name is missing.

Because of this, I recommend that the paper should be accepted for publication after a minor revision.

Author Response

Dear Editors and Reviewers:

We would like to first thank the editor and reviewers for their feedback. The reviewers’ comments are specifically addressed below, with reference to the point in the manuscript into which feedback has been incorporated. Responds to the reviewer’s comments:

Reviewer #2:

In their paper "Dynamic mechanical and creep behaviour ...", Foroughi and coworkers give an overview over some important mechanical properties of certain nanocomposite piezoelectric fibers. Even so these properties are explained in a rather qualitative and descriptive way, their knowledge is for sure of major importance for the development and characterization of nanocomposite fiber based devices and thus of interest for the readers of nanomaterials. The methodology sounds solid, and the publication continues previous work by the same authors in the field.

  1. line 73: the abbreviation "MWNT" has not been defined before.
    • The full name was written in the text.

  2. line 144: both the storage modulus as well as the loss modulus are called E'.
    • It corrected to: Storage modulus (E′), loss modulus (E″)

  3. line 152: shouldn't the unit of "probe current" be mA instead of mV?
    • Corrected to 30 nA

  4. Figure 2: The authors should provide some more details on what exactly is shown in Figure (f).
    • Added to the figure caption and text: Figure 2f shows the low content of graphene nanosheets with the dimension of 200-400 nm

  5. Figure 3: in both figures (a) and (b) the x-axis label should be "Temperature (°C)", not "Tempreture (°C)".
    • Both Figures were corrected.

  6. Table 1: what are the error bars for Tm, Tc, and Xc? The same for the peak temperatures in line 307.
    • For Tm, Tc, and Xc the tests repeated 10 time for each sample and the measured value were the same and the amount of error was negligible.

  7. Figure 7: Shouldn't (f) also be "creep recovery"? In the caption only d and e are mentioned.

                  It is corrected for figure 7(d-f).

  1. The authors should check the references. In line 508, a "n/a" is given for year or volume. For ref 41 (line 524) the journal name is missing.
    • References were updated.

Round 2

Reviewer 1 Report

It can be published in the present form.

This manuscript is a resubmission of an earlier submission. The following is a list of the peer review reports and author responses from that submission.

Round 1

Reviewer 1 Report

Recently, poly(vinylidene fluoride) (PVDF) has been the subject of extensive research, due to the piezoelectric properties of its β crystalline form. Studies are aimed at increasing the content of β phase. In this respect, the authors of the reviewed work have already achieved great success. The composite fibers with a PVDF matrix obtained by them are characterized by a high β phase and therefore have high electro active properties.

The reviewed manuscript (ID: nanomaterials-1246294, entitled: „Dynamic mechanical and creep behaviour of meltspun PVDF nanocomposite fibers”) is a continuation of the previously published research and focuses on the study of storage modulus and creep behavior. The article also presents the study of fiber morphology using SEM microscopy and crystal structure by means of DSC. In their work, the researchers carefully documented that the addition of rGO and BT nanofillers significantly improved the creep resistance of PVDF, and showed the long-term creep performance of investigated nanocomposite fibers.

Although I recommend the article to be printed, I suggest small correction, namely:

  1. I propose to unify the colors assigned to the samples in Figure 7(d-f), as in Figure 7(a-c). This change will allow for a clearer understanding of the observed changes depending on the type of nanofiller and temperature.
  2. It also seems too simple to combine a very high recovery for PVDF/rGo fibers with their crystallinity (line 337), when this crystallinity is only 1% higher than the other tested fibers. Perhaps there are other factors to consider as well.

Reviewer 2 Report

This study has little novelty. It looks like a simple summary of the authors' previous works (Refs. 11, 14, 15). The most unacceptable issue is that in the analysis of the Dynamic mechanical behavior (3.3) and Tensile creep behavior (3.4), many results are quoted from references. For example: This can be attributed to reinforcing effect [30] of rGO.Sheets of rGO 269 have high aspect ratio with strong interfacial adhesion for the integration in the PVDF matrix.[31]

Based on theses reasons, I donot suggest the acceptance of this manuscript.

Reviewer 3 Report

In this research titled “Dynamic mechanical and creep behaviour of 2 meltspun PVDF nanocomposite fibers”, The meltspun PVDF nanocomposite fibers were fabricated and characterized. The mechanical characterization part of the manuscript have been organized good but it cannot be published in this format. My main comments were listed below:

1- The novelty and application of rGO-BT-PVDF fibers should be addressed in the manuscript clearly.

2- The synthesis methods of GO and rGO should be mentioned in detail. As an example, Hummer’s method is used for GO not rGO. Which method was used for rGO?conditions? Chemical or thermal method?

3- This manuscript suffers from the lack of characterization for BT nanoparticles, GO and rGO nanosheets. Without them, this manuscript cannot be published.

4- More advanced interpretation should be used to explain precisely the role of nanomaterials in the mechanical properties.

5- The language and scientific terms should be checked in the whole manuscript. As an example: Hummer’s method not Hummer method or Tg (g should be subscript), etc.